# 20 Years of Indian Gamma Ray Astronomy Using Imaging Cherenkov Telescopes and Road Ahead

**Krishna Kumar Singh** [1,*] and **Kuldeep Kumar Yadav** [1,2]

1    Astrophysical Sciences Division, Bhabha Atomic Research Centre, Mumbai 400085, India;
     kkyadav@barc.gov.in
2    Homi Bhabha National Institute, Anushakti Nagar, Mumbai 400094, India
*    Correspondence: kksastro@barc.gov.in

**Abstract:** The field of ground-based $\gamma$-ray astronomy has made very significant advances over the last three decades with the extremely successful operations of several atmospheric Cherenkov telescopes worldwide. The advent of the imaging Cherenkov technique for indirect detection of cosmic $\gamma$ rays has immensely contributed to this field with the discovery of more than 220 $\gamma$-ray sources in the Universe. This has greatly improved our understanding of the various astrophysical processes involved in the non-thermal emission at energies above 100 GeV. In this paper, we summarize the important results achieved by the Indian $\gamma$-ray astronomers from the GeV-TeV observations using imaging Cherenkov telescopes over the last two decades. We mainly emphasize the results obtained from the observations of active galactic nuclei with the *TACTIC* (TeV Atmospheric Cherenkov Telescope with Imaging Camera) telescope, which has been operational since 1997 at Mount Abu, India. We also discuss the future plans of the Indian $\gamma$-ray astronomy program with special focus on the scientific objectives of the recently installed 21 m diameter MACE (Major Atmospheric Cherenkov Experiment) telescope at Hanle, India.

**Keywords:** gamma ray astronomy; imaging atmospheric Cherenkov technique; TeV gamma-rays; non-thermal radiation

## 1. Introduction

The concept of gamma-ray astronomy (GRA) was first coined by Phillip Morrison in 1958 [1]. In a seminal paper, Morrison suggested that nuclear or high-energy processes give rise to continuous and discrete spectra of $\gamma$-radiation over the energy range of 0.2–400 MeV in astronomical objects. According to Morrison, continuum $\gamma$-radiation from astronomical sources is produced by three physical processes: synchrotron radiation, bremsstrahlung or braking radiation, and radiative decay of neutral pions, whereas discrete $\gamma$-ray lines can be emitted by de-excitation of nuclei formed by radioactive decay and electron-positron pair annihilation. These $\gamma$-ray photons are directly related to their origins and relatively accessible to observation. Crab Nebula, extragalactic radio source M87, and Cygnus A were suggested to be possible sources of the cosmic $\gamma$-radiation. In 1960, Cocconi proposed the possibility of detecting the TeV $\gamma$-ray photons from discrete astronomical sources using an air shower telescope [2]. This motivated the Crimean group to develop a ground-based experimental technique for recording the extensive air showers initiated by the TeV $\gamma$-ray photons by detecting the Cherenkov radiation produced from them in the Earth's atmosphere [3]. Using this experiment, Crab Nebula, radio galaxies, and supernova remnants were observed by the Crimean group.

The first telescope for ground-based GRA using the atmospheric Cherenkov technique was constructed by G. G. Fazio in 1968. The Whipple telescope with a 10 m optical reflector reported the first tentative detection of cosmic $\gamma$ rays with energy above 250 GeV in 1972 [4]. The first detection of cosmic $\gamma$-radiation with the space-based telescope Explorer XI was demonstrated in 1965 [5]. By this period, it was thus realized that cosmic $\gamma$-ray photons

can be either observed directly by space-based detectors onboard satellites/balloons in the stratosphere or indirectly from the ground by detecting the extensive air showers produced by the $\gamma$-ray photons. Space-based GRA was developed to cover the MeV-GeV energy range, whereas ground-based GRA explores the GeV-TeV energy band. In 1989, the atmospheric Cherenkov imaging technique was applied to discriminate between images of cosmic $\gamma$ ray- and proton-induced showers based on their shape and orientation at the focal plane of the Whipple telescope. This technique offered the first detection of $\gamma$ rays with energy above 700 GeV from the Crab Nebula at very high statistical significance level [6]. With an excellent capability of rejecting the cosmic ray hadronic background, a large field of view and very good energy resolution, imaging atmospheric Cherenkov telescopes (IACTs) laid a strong foundation for the present and future ground-based GRA in GeV-TeV energy range.

In India, ground-based experimental investigations for detecting celestial TeV $\gamma$ rays were initiated in 1969 by setting up an atmospheric Cherenkov telescope with two search light mirrors mounted on an orienting platform at Ooty (altitude $\sim$ 2300 m) in south India, more or less concurrently with the corresponding efforts made worldwide. Over the years, the number of mirrors was increased, and detection method were improved. For 16 years, an array of tracking Cherenkov telescopes was operated in and around Ooty for observing TeV $\gamma$-ray emission from radio pulsars. In order to take advantage of better sky conditions, the experimental set-up was shifted from Ooty to Pachmarhi (altitude $\sim$ 1075 m) in central India in 1986. Another Indian group started observing the night sky by setting up an atmospheric fluorescence experiment at Gulmarg (altitude $\sim$ 2700 m) in North India using two bare-faced photomultiplier tubes separated by 1 m in 1972. Later on, an experimental set-up consisting of six equatorially mounted parabolic search light mirrors, similar to the Pachmarhi facility, was also installed at Gulmarg in 1984 [7]. This was the first generation of indigenously developed multi-mirror atmospheric Cherenkov telescope commissioned at Gulmarg for dedicated GRA in the TeV energy range. Surprisingly, the mirrors for Gulmarg Cherenkov telescope were found in the Mumbai junk market. These parabolic mirrors were searchlight reflectors of the Second World War vintage, and evidently without great optical quality. A special design feature of this telescope was that two identical sections of the telescope could be deployed for observing either two different sources concurrently, or, taking observations on the same source, with one section viewing on-source and the other engaged in the simultaneous off-source monitoring. The later mode was suitable for observations of $\gamma$-ray sources with episodic emissions. This facility was operated during 1984–1989. A multi-station wide-angle photomultiplier-tubes-based experiment was also set up simultaneously at Gulmarg and Srinagar (altitude $\sim$ 1500 m) to search for simultaneous arrival of short timescale $\gamma$-ray and optical emissions from the explosive evaporation of primordial black holes [8–10]. This experiment employed three closely placed, vertically oriented, large-area photomultiplier tubes at a baseline of 30 km between Gulmarg and Srinagar. The most promising result from this experiment was based on a time-series analysis of atmospheric Cherenkov pulses recorded in the direction of Cygnus X-3.

Equipped with over twenty years of experience in experimental techniques and a good understanding of the theoretical concepts in GRA, researchers associated with the Gulmarg observatory moved to Mount Abu (altitude $\sim$ 1300 m) in the western region of India in 1992 for setting up a new GRA facility with higher detection sensitivity. The first IACT in Asia, TACTIC (TeV Atmospheric Cherenkov Telescope with Imaging Camera), was installed at Mount Abu in 1997 [11]. In order to reduce the energy threshold to a few tens of GeV, the Pachmarhi group shifted their activity to a high-altitude site with low night sky background at Hanle (altitude $\sim$ 4200 m) in the Himalayas. A seven-element wavefront sampling telescope, HAGAR (High Altitude GAmma Ray telescope) was commissioned at Hanle in 2008 [12]. Another IACT, namely MACE (Major Atmospheric Cherenkov Experiment), has recently been installed at Hanle by a collaboration of Indian gamma ray astronomers called Himalayan Gamma Ray Observatory (HiGRO) [13].

Significant advances in the instrumentation and technology have led to the development of several state-of-the-art IACTs around the globe in the last 30 years and very exciting

results have been produced by them in the $\gamma$-ray energy range above $\sim$100 GeV [14–20]. The motivation behind GRA using IACTs is to explore the most violent and energetic physical processes for addressing the questions related to the origin of the GeV-TeV cosmic $\gamma$ rays and radiative and acceleration processes under extreme physical conditions in the non-thermal Universe. The field of GRA is also expected to make an impact on cosmology and astroparticle physics through the propagation effects of $\gamma$ rays over cosmological distances. Unveiling the nature of dark matter candidates such as weakly interacting massive particles, axion-like particles, comprehension of the astrophysical acceleration processes and lepto-hadronic origin of high $\gamma$ rays are the major challenges for the present and future instruments for the ground-based GRA research. In this contribution, we present the scientific achievements of the Indian GRA program using IACTs in the last twenty years and future roadmap for the next decade. The structure of paper is as follows. In Section 2, important results from early times of an Indian GRA program are briefly described. Current status of the experimental facilities and results obtained so far are discussed in Sections 3 and 4, respectively. Section 5 outlines the future program of Indian GRA with IACT. Finally, we conclude in Section 6.

## 2. Early Times: Important Achievements

Ground-based GRA was pioneered in India in 1969, soon after the discovery of a pulsar in the Crab Nebula in 1968. In the beginning, a very systematic search was made for pulsed $\gamma$-ray emission above the detection threshold energy of $\sim$10 TeV from six pulsars using Cherenkov telescopes at Ooty [21]. Not having found any statistically significant TeV $\gamma$-ray emission from any source, upper limits of the order of $10^{-11}$ ph cm$^{-2}$ s$^{-1}$ were placed on the $\gamma$-ray fluxes [22]. A number of isolated pulsars (Crab, Vela, PSR0355+54, Geminga) and X-ray binaries (Hercules X-1) were monitored by experimental set-ups at Ooty and Pachmarhi, and positive signals were detected from them on several occasions. The Ooty array started the observation of the Crab Pulsar in 1977 and signals in the time-averaged phasograms were detected only on a few occasions. The Durham University group detected pulsed TeV $\gamma$-ray emission from the Crab pulsar during two transients lasting 15 min, each in October 1981 [23], while the Ooty group reported that the transient duration could be on time scales of minutes or even seconds [24]. A 15 min interval was identified on 23 January 1985, during which pulsed TeV $\gamma$ rays were detected at $5.1\sigma$ confidence level from the Crab pulsar [25]. An important feature of this observation was that two independent telescopes tracking the Crab pulsar from locations at Ooty separated by 11 km detected the signal, while a third telescope adjacent to one of them but pointing towards a background region did not show any effect. This strengthened the inference of a transient pulsed TeV $\gamma$-ray emission with the peak of the pulse at the position of radio main-pulse. An identical burst from the Crab pulsar was detected at $6\sigma$ statistical significance level on 2 January 1989 while operating at Pachmarhi with an altogether different set-up [26]. This burst was observed by all five quasi-independent telescopes and lasted for 5 min. Evidence for variability over time scales of minutes and possibly hours in the TeV light curves of the Crab pulsar was also investigated [27].

The observations on Vela pulsar were made between February and March 1979 for 17 nights by the Ooty observatory, and detection of $\gamma$ rays at energies above 500 GeV was reported from the Vela pulsar [28]. This observation detected two narrow peaks separated by 0.42 in phase space (a characteristic feature noticed at lower energy by satellite-based observations) but did not provide absolute phase information. Data from later observations provided information on the absolute phase. The signal-to-noise ratio improved significantly when lower energy events were preferentially selected, and the resultant TeV $\gamma$-ray phasogram of Vela pulsar showed a $4\sigma$ peak aligned with the optical first pulse position [29]. A weak second pulse separated by 0.43 from the main pulse in the phase space was also noticed at $1.5\sigma$ level. From the excesses observed at different $\gamma$-ray energy thresholds (from 4.9 teV to 10.4 TeV) over a span of five years, the integral energy spectrum was found to follow a power-law with slope $-(2.5 \pm 0.3)$. The radio pulsar PSR0355+54 was observed

in December 1987 for 25 h in the TeV energy range at Pachmarhi, and a steady pulsed emission signal at a phase of 0.53 with respect to the radio pulse was detected at $4.3\sigma$ level above 1.3 TeV energy threshold [30]. A large increase in the trigger rate in the direction of X-ray binary candidate Hercules X-1 was observed in the atmospheric Cherenkov telescope array at Pachmarhi on 11 April 1986. The accidental coincidence rate did not show any increase during this burst and the number of detected $\gamma$ ray events amounted to $\sim$54% of the cosmic-ray flux, resulting in a $42\sigma$ effect [31]. This was the largest TeV $\gamma$ ray signal from any source until that time. In the search for pulsed TeV gamma rays in the archival data of 1984–1985 on Geming pulsar while operating at Ooty, two peaks at phases 0.4 and 0.9 (as observed in COS-B satellite data) with a separation of 0.5 in phase were seen during a few minutes of short term activity of the source [32].

The modest start in the then-budding field of GRA at Gulmarg was made in 1972 by setting up an atmospheric scintillation experiment to search for prompt $\gamma$-ray emissions from supernovae explosions and primordial black hole outbursts. In a supplementary mode of operations, this experiment was deployed as the atmospheric Cherenkov detection technique to investigate the cosmic-ray energy spectrum around the *knee* position and to obtain corroborative evidence for ultra-high-energy $\gamma$-ray emission from the binary system Cygnus X-3 during the exploratory stage between 1974 and 1984. The energy threshold of the system for showers initiated by primary cosmic rays was determined to be $\sim$500 TeV from the fluctuations in the night-sky background [33]. A pulse-height analysis of the Cherenkov pulses detected by the wide-angle photomultiplier tube system at Gulmarg indicated a break near $10^{15}$ eV in the cosmic-ray spectrum, and it was argued to be primary in nature [33]. In the drift scan mode (telescope is kept stationary and sky is scanned as the Earth rotates), the Gulmarg experimental facility detected Cherenkov pulses during 1976–1978 corresponding to an average event rate of 55 $h^{-1}$. During this ground-based search for episodic cosmic events at Gulmarg, the arrival time distribution of atmospheric Cherenkov pulses revealed a significant overabundance of cosmic ray events inter-separation of $<$40 s [34]. Analysis of events recorded during 1976–1977 indicated the presence of a $4.5\sigma$ significant phase-dependent signal from the binary system Cygnus X-3 with the characteristic modulation period of 4.8 h [35]. This was the most important result from the Gulmarg wide-angle photomultiplier tube experiment based on a time-series analysis of the Cherenkov pulses. Following its publication [35], the Gulmarg result on Cygnus X-3 attracted a lot of international attention. Evidence for possible TeV $\gamma$-ray emission from several candidate sources were obtained during the observation period 1985–1990 with the multi-mirror atmospheric Cherenkov telescope at Gulmarg [36]. A possible discovery result from these observations was the detection of TeV $\gamma$-ray signal from the prototype cataclysmic variable AM-Herculis [37]. Other interesting results with the Gulmarg telescope were obtained from the observations of pulsar PSR 0355+54 [38], Crab nebula/pulsar [39], X-ray binary source Cassiopeia $\gamma$-1 [40], and search for millisecond pulsar in Cygnus X-3 [41].

The real breakthrough in ground-based GRA occurred in 1989, when the Whipple Collaboration used Hillas image parameters (proposed by A. M. Hillas in 1985 [42]) to distinguish very effectively between background hadronic showers and TeV gamma ray showers from a point source [6]. Subsequently, development of several IACTs was initiated throughout the world for GRA in the GeV-TeV energy range. The early leads from the Gulmarg exploratory phase and contemporary global trends in the field of GRA provided a novel approach and motivation to the researchers associated with the Gulmarg observatory for setting up lower-threshold energy $\gamma$-ray telescopes like TACTIC and MACE based on the imaging atmospheric Cherenkov technique. The TACTIC telescope belongs to first-generation IACTs such as Whipple, HEGRA, CANGAROO, SHALON, and CAT, whereas MACE can be placed among the second-generation state-of-the-art telescopes like H.E.S.S., VERITAS, and MAGIC on the world map.

### 3. TACTIC Telescope

The TACTIC telescope was set up at Mount Abu (24.6° N, 72.7° E, 1300 m above sea level) in 1997. A comprehensive site survey was performed in 1993, and Gurushikhar in Mount Abu turned out to be the best-known location for GRA research using imaging the Cherenkov technique [43]. This site offered a very significant enhancement in the observation time (~1200 h per year) more or less evenly spaced throughout the year with respect to the Gulmarg observatory. Mount Abu is a hill resort with good logistics, mild climate, dust-free atmosphere and offers ready accessibility to major Indian cities. The site is located at nearly the same longitudinal belt in which several major astronomical experiments in India were operational during that period. This fortuitous longitude clustering was greatly helpful in time-coordinated multi-band observations on candidate $\gamma$-ray sources. The longitude of the Mount Abu site is also important in long-term monitoring of the $\gamma$-ray sources as compared to other GRA observatories around the world.

The design of TACTIC telescope is based on imaging atmospheric Cherenkov technique for indirect detection of TeV photons from the cosmic sources [11]. A photograph of the observatory at Mount Abu is depicted in Figure 1. The instrument at the center of the array has been deployed as an imaging unit for TeV $\gamma$-ray observations since 1997. The imaging element is equipped with an altitude-azimuth mounted light collector of ~4.0 m diameter and ~3.8 m focal length. The light collector employs 34 front facing. aluminum-coated, glass spherical mirrors of 0.6 m diameter, each providing a light collection area of ~10 m². When all the 34 mirror-facets are properly aligned on the basket, the overall light reflector corresponds to a quasi-paraboloid surface. With focal length-to-diameter ratio ~1, hybrid design of light collector is close to the Davies-Cotton design. This was achieved by deploying the shorter focal length mirror facets close to the principal axis of basket, while mirrors with longer focal lengths are placed around the periphery using longer studs on the frame structure to raise their pole position. This arrangement minimizes the off-axis effects on overall spot size of the reflector. A maximum spot size of ~4 arcmin can be expected in the image plane of the telescope. The imaging camera at the focal plane uses an array of photomultiplier tubes (pixels) to detect the Cherenkov light flash with a resolution of 0.31°. Initially, source observations using TACTIC started with only 81-pixel imaging camera covering a field of view of ~2.8° × 2.8° in early 1997. Within a few days of its first light in 1997, the telescope with its prototype 81-pixel camera successfully detected a flaring activity from the blazar Mrk 501 during April–May 1997. This observation was almost synchronized by five other gamma-ray telescopes operating around the globe [44]. The prototype camera was first upgraded to 144 pixels, and the final camera configuration of 349-pixels with a field of view of ~6° × 6° was attained in December 2000. The simulation studies using CORSIKA [45] suggested threshold energy of the TACTIC imaging telescope for $\gamma$-rays and protons to be ~1.0 TeV and ~1.8 TeV, respectively. The sensitivity of the telescope was estimated as detection of $5\sigma$ steady signal from the standard candle Crab Nebula in 25 h. A detailed description of the TACTIC telescope design and instrumentation can be found in [46–51]. The data recorded by the TACTIC telescope are first corrected for inter-pixel gain variation and then subjected to the standard image cleaning procedure [52]. The image cleaning threshold levels (for boundary and core pixels) are optimized on the Crab Nebula data. The clean images are characterized by calculating their *Hillas parameters* [42] followed by the application of standard Dynamic Supercuts procedure [53] to segregate $\gamma$-ray like events from the huge cosmic-ray background events. The significance $\gamma$-ray like excess events is estimated using the maximum-likelihood ratio method proposed by Li and Ma [54].

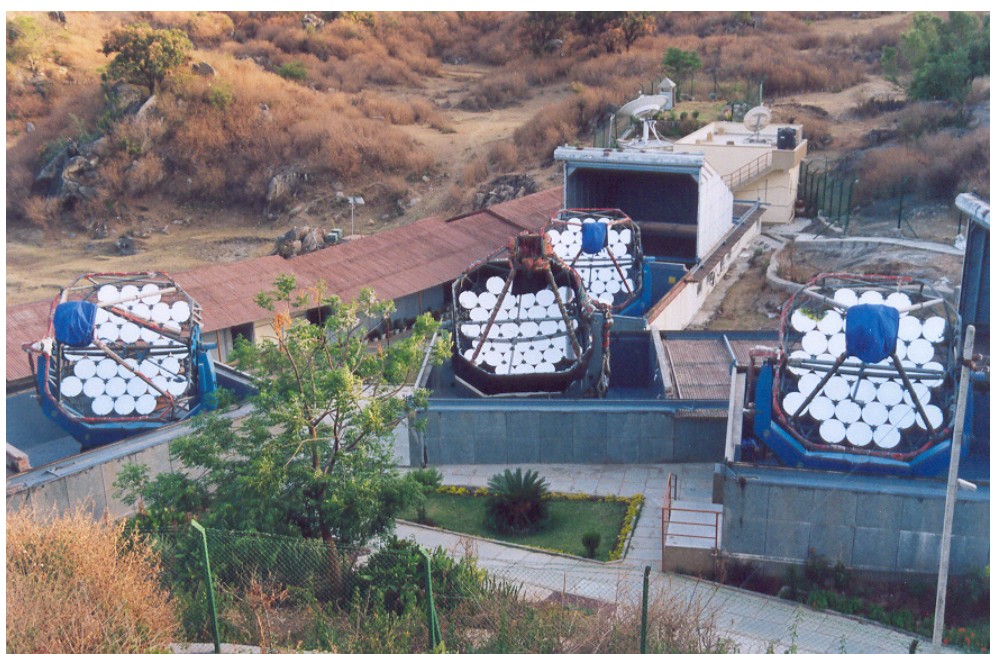

**Figure 1.** The TeV Atmospheric Cherenkov Telescope with Imaging Camera (TACTIC) telescope array operational since 2001 at Mount Abu, India. Single telescope at the center is being used as an imaging unit for TeV $\gamma$-ray observations.

A major upgrade program was taken up in 2011 to improve the overall performance of the telescope. The main motivation for the hardware and software upgrade of the system was to increase its sensitivity and lower the threshold energy. This translated to the reduction in the threshold energy of the telescope for cosmic rays from 1.8 TeV to 1.4 TeV and from 1.2 TeV to 0.8 TeV for the $\gamma$-ray events [55,56]. Application of gamma/hadron separation strategies using artificial neural networks and random forest classification further enhanced the performance of the telescope after upgrade [57,58]. The upgraded telescope has an improved sensitivity to detect the TeV $\gamma$-ray emission from the Crab Nebula at $5\sigma$ statistical significance level in an observation time of 12 h as compared to 25 h earlier. The $\gamma$-ray detection rate from the Crab Nebula also increased from $\sim$10 h$^{-1}$ to $\sim$15 h$^{-1}$ (defined as one Crab Unit for TACTIC). The TACTIC telescope with significant improvement in its performance has greatly helped in the monitoring of potential $\gamma$-ray sources in multi-TeV energy range during flaring activities for a short period and in low state for a long duration.

## 4. Important Results from TACTIC Observations

The TACTIC telescope started regular observations of TeV $\gamma$-ray sources with a full 349-pixel camera in January 2001. First observations were carried out on the Crab Nebula for 41.5 h during 19 January–23 February 2001. A statistically significant number of $\gamma$-ray like events ($447 \pm 71$) were detected from the direction of Crab Nebula at $6.3\sigma$ significance level. A long-term monitoring of the Crab Nebula with TACTIC between 2003 and 2010 (before upgrade) yielded ($3742 \pm 192$) $\gamma$-ray like events with statistical significance of $\sim$20$\sigma$ in $\sim$400 h of live observation time [56]. Results from this observation are shown in Figure 2a–c. The differential energy spectrum of the Crab Nebula using the consolidated TACTIC data was very well described by a power law of the form

$$\frac{d\phi}{dE} \;=\; f_0 \left(\frac{E}{1\text{TeV}}\right)^{-\Gamma} \tag{1}$$

where $f_0 = (2.66 \pm 0.29) \times 10^{-11}$ ph cm$^{-2}$ s$^{-1}$ TeV$^{-1}$ is the normalization constant at energy E = 1 TeV and $\Gamma = 2.56 \pm 0.10$ is the photon spectral index [56]. The spectrum obtained

matches reasonably well with that measured by the Whipple and HEGRA groups [59,60]. Use of artificial neural network methodology for energy reconstruction of $\gamma$ ray events detected by the TACTIC telescope was found to be more effective at higher energies and led to determining the Crab Nebula energy spectrum in the energy range 1–24 TeV [61,62]. The TACTIC telescope had been mainly deployed for observations of blazars, which represent the dominant population of TeV $\gamma$-ray sources in the extragalactic Universe. Blazars are radio-loud active galactic nuclei (AGN) having an elliptical host galaxy with a supermassive black hole at the center. Broadband non-thermal emission over the entire electromagnetic spectrum ranging from radio to TeV $\gamma$-rays is produced from blazars in a relativistic plasma jet originating from the central region and oriented at small angles to our line of sight. A detailed description of the present understanding of blazars is given in [63,64]. Important results obtained from the TACTIC observations of the potential TeV $\gamma$-ray sources over the last 20 years are described in the following subsections.

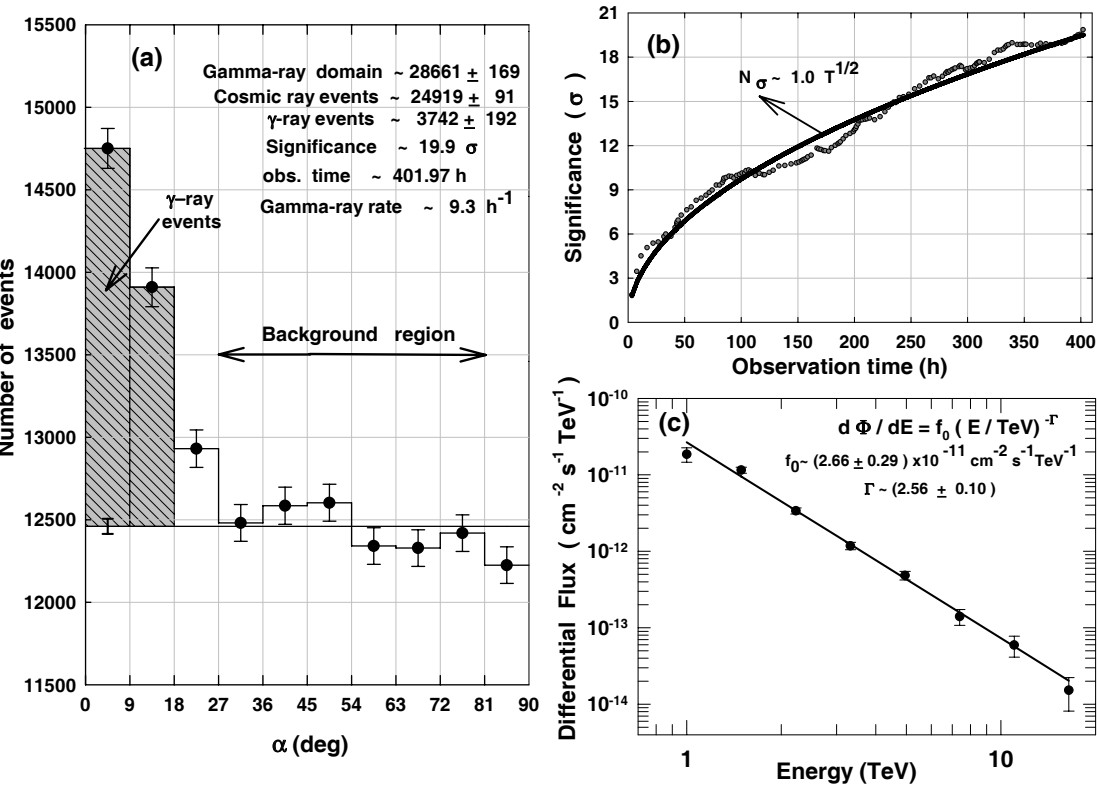

**Figure 2.** (**a**) TeV $\gamma$-ray detection from Crab Nebula with the TACTIC telescope; (**b**) Cumulative significance level as a function of observation time; and (**c**) Differential energy spectrum of the Crab Nebula measured by TACTIC [56]. Reprinted by permission from Springer Nature Customer Service Centre GmbH: Springer Nature Pramana, Long- term performance evaluation of the TACTIC imaging telescope using ∼400 h Crab Nebula observation during 2003–2010, A K Tickoo et al., Copyright 2014.

### 4.1. Mrk 501

Mrk 501 is one of the brightest TeV $\gamma$-ray blazars at redshift $z = 0.034$ in the extragalactic Universe. The first TeV emission from this source was detected in 1995 by the Whipple telescope above 0.3 TeV [65]. During April–May 1997, a significant detection of $\gamma$-ray events from this active galaxy was claimed by the newly-commissioned TACTIC telescope at statistical significance of ∼13$\sigma$ in ∼50 h of observation time. This observation with the TACTIC telescope produced evidence for a series of TeV $\gamma$-ray flares from Mrk 501 [44]. The TeV $\gamma$-ray emission from this source was further monitored by the TACTIC telescope during March–May 2005 and February–May 2006 for ∼46 h and ∼67 h, respectively, [66,67]. During 2005 observations, no significant $\gamma$-ray emission was detected from the source direc-

tion, and therefore an upper limit of $4.62 \times 10^{-12}$ ph cm$^{-2}$ s$^{-1}$ at 3$\sigma$ confidence level was placed on the integrated flux above 1 TeV. However, during 2006 observations, presence of a TeV $\gamma$-ray signal from the source with a statistical significance of 7.5$\sigma$ was found. The time-averaged differential energy spectrum of the source was described by a power law in the energy range 1–11 TeV with $f_0 = (1.66 \pm 0.52) \times 10^{-11}$ ph cm$^{-2}$ s$^{-1}$ TeV$^{-1}$ and $\Gamma = 2.80 \pm 0.27$. These results closely followed those obtained by the HEGRA collaboration during 1998–1999, except for the exponential cutoff feature in the spectrum at 2.61 TeV [68].

A multi-wavelength study of the TeV $\gamma$-ray emission from Mrk 501 was performed using TACTIC observations during April–May, 2012 [69]. Analysis of the TACTIC light curve during this period indicated a relatively high gamma-ray emission state of the source between 22 and 27 May 2012. The time-averaged differential energy spectrum measured by TACTIC during the high-state of Mrk 501 is shown in Figure 3. The intrinsic TeV $\gamma$-ray spectrum was estimated from the observed spectra after correcting for absorption due to extragalactic background light (EBL) as explained in [70,71]. The intrinsic emission from the source can be described by a power law with $f_0 = (2.73 \pm 0.51) \times 10^{-11}$ ph cm$^{-2}$ s$^{-1}$ TeV$^{-1}$ and $\Gamma = 2.19 \pm 0.18$ in the energy range 0.85–17 TeV. This TeV $\gamma$-ray emission was satisfactorily reproduced by the synchrotron self-Compton process under the framework of widely used homogeneous single zone leptonic model for broadband spectral energy distribution of blazars [72]. The model predicted a tangled magnetic field of 0.12 Gauss in a spherical emission region of radius $6.1 \times 10^{15}$ cm. The energy distribution of relativistic electrons in the emission zone followed a smooth broken power law with indices 2.1 and 4.9 before and after the break, respectively [69,73]. These findings are found to be broadly consistent with the results from the multi-wavelength observations of the source during April–August, 2013, including data from MAGIC and VERITAS telescopes [74].

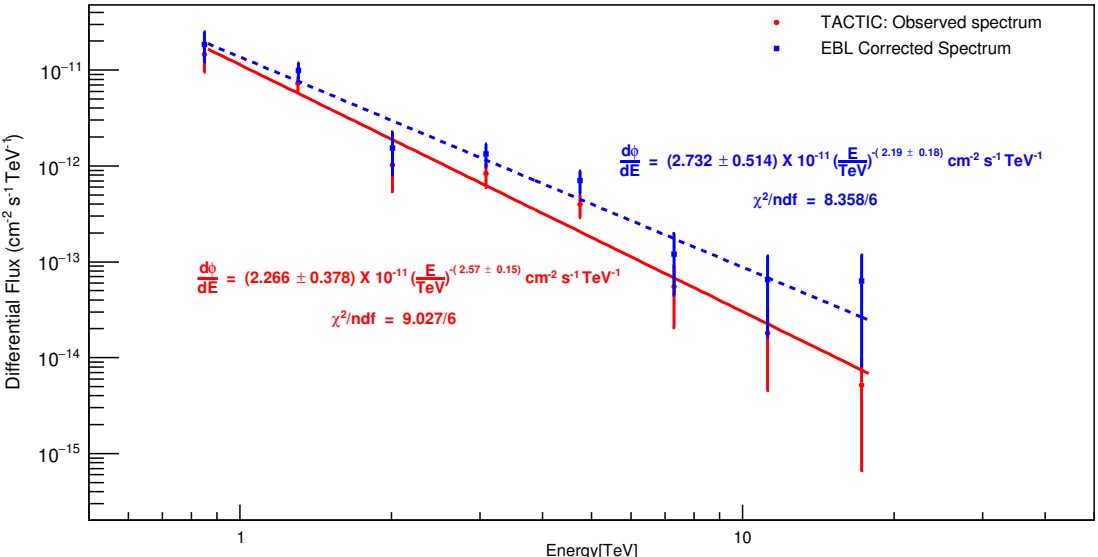

**Figure 3.** Observed and intrinsic differential energy spectra of Mrk 501 during high state from 22 to 27 May 2012 measured by the TACTIC telescope. Reproduced with permission from the journal, Reference [69], Copyright 2017, Elsevier.

Mrk 501 showed a major TeV $\gamma$-ray flaring activity on the night of 23–24 June 2014 with a flux variation characterized by the flux doubling timescale of a few minutes [75]. Unfortunately, the TACTIC telescope could not observe Mrk 501 during this flaring episode due to bad weather conditions. However, a near-simultaneous multi-wavelength study of this giant flaring episode by us suggested a correlation between the TeV $\gamma$-ray emission and soft X-ray emission [76]. The soft X-ray photon spectral index was observed to be anti-correlated with the integral flux showing harder-when-brighter behavior. The nature of X-ray and $\gamma$-ray emissions from Mrk 501 shows different behavior during low and high activity states of the source [63]. Therefore, further monitoring of Mrk 501 is very important

to explore the exact nature of the TeV $\gamma$-ray emission, and TACTIC will continue to observe this source as one of its potential targets.

### 4.2. 1ES 2344+514

First TeV $\gamma$-ray emission from the blazar 1ES 2344+514 ($z = 0.044$) was discovered by Whipple Collaboration in 1995. The TACTIC telescope monitored this source from 18 October to 9 December 2004 and 27 October 2005 to 1 January 2006 for a total live time of $\sim$60 h in the zenith angle range of 27°–45° [66,77]. Analysis of the data indicated absence of a statistically significant TeV $\gamma$-ray signal from the source direction. Therefore, an upper limit of $3.84 \times 10^{-12}$ ph cm$^{-2}$ s$^{-1}$ at $3\sigma$ confidence level on the integral flux above 1.5 TeV was estimated. The derived upper limit from TACTIC observations was in agreement with the detection of very high-energy $\gamma$-ray emission in low emission state of the source during 2005–2006 with the MAGIC telescope [78].

### 4.3. H 1426+428

H 1486+428 was discovered in X-ray observations at redshift $z = 0.129$. Whipple group reported the first TeV $\gamma$-ray detection from this blazar in 2002 at a high statistical significance level using the data collected during 1999–2001. The TACTIC telescope observed H 1486+428 for 244 h between March 2004 and June 2007 in the continuous source tracking mode over the zenith angle range of 18°–45° [79,80]. The TeV $\gamma$-ray emission from this source was found to be below the TACTIC sensitivity level during the period of these observations. In the absence of statistically significant detection of the TeV $\gamma$-ray signal from the source detection, an upper limit of $1.18 \times 10^{-12}$ ph cm$^{-2}$ s$^{-1}$ ($\sim$13% of the TACTIC detected Crab Nebula integrated $\gamma$-ray flux above 1 TeV) at $3\sigma$ confidence level was placed on the integral flux from the source. The TACTIC results on the source H 1426+428 were consistent with those obtained with the CELESTE system during the same period [81] but in conflict with GT-48 telescope observations during the period 15–25 April 2004, wherein a $\gamma$-ray signal at $5.8\sigma$ statistical significance level was reported [82].

### 4.4. Mrk 421

Mrk 421 at redshift $z = 0.031$ is the first blazar detected at TeV energies by ground-based IACT, and the second source after the Crab Nebula. The pioneer Whipple telescope discovered $\gamma$-ray emission above 0.5 TeV from Mrk 421 in 1992. The TACTIC telescope was deployed to monitor the TeV $\gamma$-ray emission from this source for the first time during April-May 1997 for 26 h. No significant detection was found, and the null result was used to estimate a $3\sigma$ confidence level upper limit of $\sim$5.0 $\times 10^{-12}$ ph cm$^{-2}$ s$^{-1}$ on the integral flux above 2 TeV [48]. This comparatively low state of the source was consistent with the time-averaged flux estimate from contemporaneous Whipple observations above 0.3 TeV. This reassured the satisfactory performance of the TACTIC telescope during the maiden observation campaign in 1997 soon after its commissioning with 81-pixel prototype imaging camera. Equipped with a full 349-pixel camera, the telescope was deployed to observe Mrk 421 during January–April, 2004. An evidence of the TeV $\gamma$-ray signal from the source direction with a statistical significance of $6.8\sigma$ in $\sim$79 h was found [83]. The differential energy spectrum of the detected $\gamma$-ray photons was derived as a power law with $\Gamma = 2.80 \pm 0.20$ in the energy range 2–9 TeV. A long-term monitoring of Mrk 421 with TACTIC was undertaken between December 2005 and April 2006 for a total observation time of $\sim$202 h [84]. A flaring activity was detected between December 2005 and February 2006. During the flaring period, presence of a strong $\gamma$-ray signal at $12\sigma$ statistical significance level in 97 h was found from the source direction. The time-averaged differential energy spectrum during the high-activity state was reasonably fitted by a power law with $f_0 = (4.66 \pm 0.46) \times 10^{-11}$ ph cm$^{-2}$ s$^{-1}$ TeV$^{-1}$ and $\Gamma = 3.11 \pm 0.11$ in the energy range 1–11 TeV [84]. Another flaring activity was detected between January–May, 2008, by the TACTIC telescope during the long-term observations of the blazar Mrk 421 from December 2006 to May 2008 [85]. The time-averaged differential spectrum during the high

emission state period was again described by a power law with $f_0 = (6.8 \pm 1.4) \times 10^{-11}$ ph cm$^{-2}$ s$^{-1}$ TeV$^{-1}$ and $\Gamma = 3.32 \pm 0.22$ in the energy band 1–10 TeV. Few important results derived from the giant flaring episodes of Mrk 421 using the TACTIC telescope are briefly discussed below.

### 4.4.1. February 2010 Giant Flare

In February 2010, a giant flaring activity was observed from Mrk 421 at all energies by almost all instruments worldwide [86]. The TACTIC telescope was engaged in the long-term monitoring of TeV $\gamma$-ray emission from the source during November 2009–May 2010, and $\sim$265 h of data were collected [87]. Clean data of $\sim$230 h revealed the presence of a TeV $\gamma$-ray signal with a statistical significance of 12.12$\sigma$. The estimated time-averaged differential energy spectrum in the energy range 1.0–16.44 TeV was fitted well by a power law with $f_0 = (1.39 \pm 0.24) \times 10^{-11}$ ph cm$^{-2}$ s$^{-1}$ TeV$^{-1}$ and $\Gamma = 2.31 \pm 0.14$ [87]. Analysis of TACTIC data from Mrk 421 during 10–23 February 2010 resulted in detection of $737 \pm 87$ $\gamma$-ray like events corresponding to a statistical significance of 8.46$\sigma$ in nearly 48 h, and data on 16 February 2010 (MJD 55243) alone yielded $172 \pm 30$ $\gamma$-ray like events in only 4.9 h with a statistical significance of 5.92$\sigma$ [88,89]. The epoch during 15–17 February 2010 (MJD 55242–55244) was characterized as the giant TeV flaring episode with a peak on 16 February 2010 (MJD 55243) from the blazar Mrk 421. Near-simultaneous daily light curves of the source for the period 10–23 February 2010 (MJD 55237–55250) in TeV, MeV-GeV, X-ray, optical, and radio bands are shown in Figure 4. With the motivation of understanding the physics involved in this major flaring activity, variability study of light curves was performed using a temporal profile with exponential rise and decay [90]. It was observed that the variation in the one-day-averaged flux from the source during the flare is characterized by fast rise and slow decay. In addition, the TeV $\gamma$-ray flux obtained from the TACTIC observations showed a strong correlation with the X-ray flux, suggesting the former to be an outcome of the synchrotron self-Compton emission process. To model the observed X-ray and $\gamma$-ray light curves, kinetic equations describing the evolution of particle distribution in the emission region were numerically solved. The injection of particle distribution into the emission region, from the putative acceleration region, was assumed to be a time -ependent power law. The synchrotron and synchrotron self-Compton emission from the evolving particle distribution in the emission region were used to reproduce the X-ray and $\gamma$-ray flares successfully. This suggested that the flaring activity of Mrk 421 could be an outcome of an efficient acceleration process associated with the increase in underlying non-thermal particle distribution [90].

### 4.4.2. High Activity in March 2012

A high flux state of the blazar Mrk 421 was observed at TeV energies between 15 and 26 March 2012 by the TACTIC telescope with the detection of $529 \pm 76$ $\gamma$-rays at 6.9$\sigma$ significance level in $\sim$39 h [91]. This translates to an event rate of $\sim$15 h$^{-1}$. The observed differential energy spectrum was described by a power law with $f_0 = (2.47 \pm 0.48) \times 10^{-11}$ ph cm$^{-2}$ s$^{-1}$ TeV$^{-1}$ and $\Gamma = 2.58 \pm 0.22$ in the energy range 0.85–9.36 TeV. The contemporaneous spectrum of the source measured by the Large Area Telescope (LAT) onboard the *Fermi* satellite in the energy range 0.1–300 GeV was also fitted well using a power law function with $f_0 = (2.14 \pm 0.26) \times 10^{-5}$ ph cm$^{-2}$ s$^{-1}$ TeV$^{-1}$ and $\Gamma = 1.81 \pm 0.07$. These two $\gamma$-ray spectra of Mrk 421 are shown in Figure 5. A prominent change in the differential energy spectral index from GeV to TeV energies was observed. This spectral break in the $\gamma$-ray spectrum might be attributed to Klein–Nishina effect on inverse Compton scattering of synchrotron photons in the TeV energy range [91].

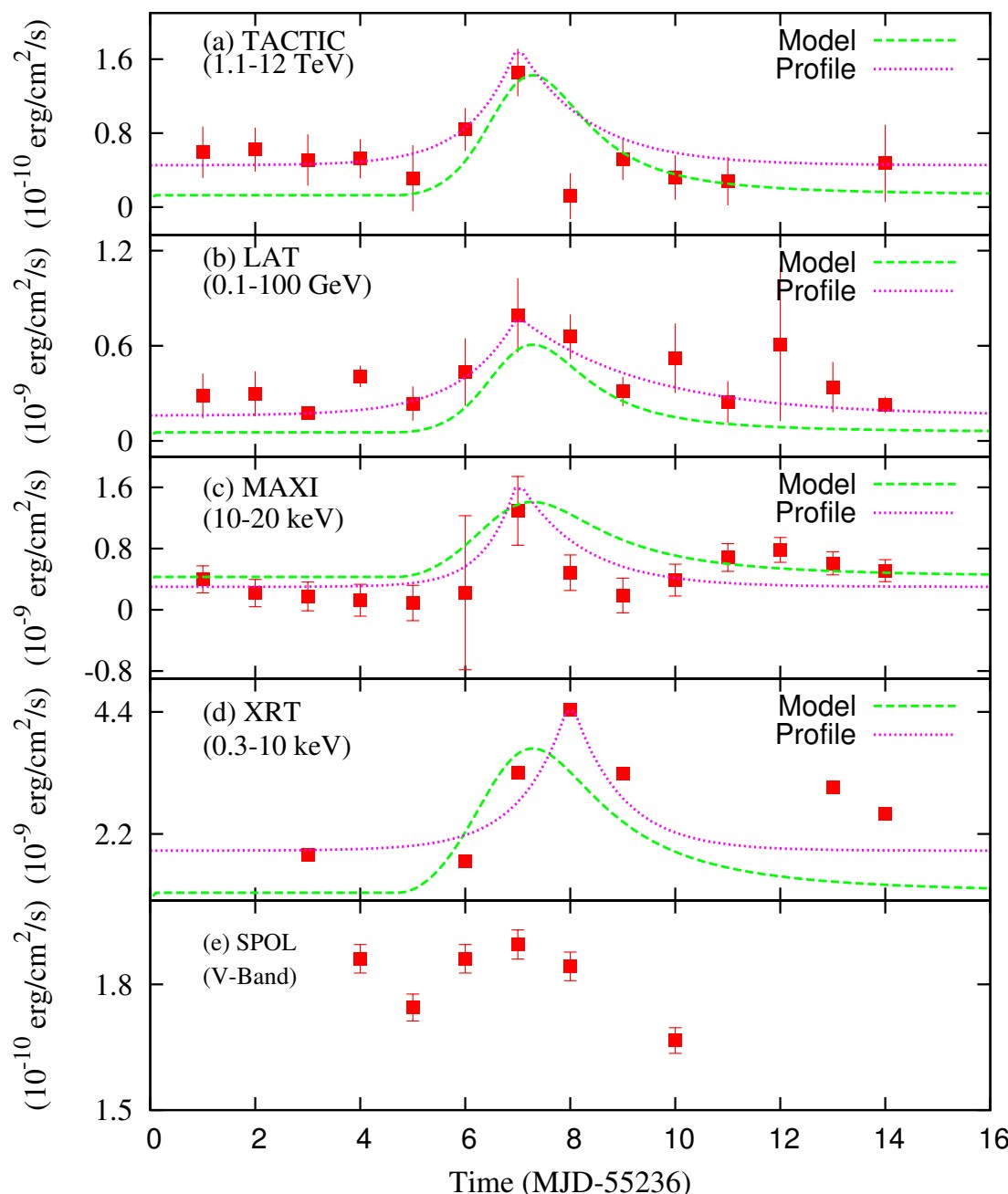

**Figure 4.** Multi-wavelength light curves of the blazar Mrk 421 during 10–23 February 2010 from various ground and space-based observations. The dotted magenta curves correspond to the best-fit temporal profile with an exponential rise and decay, while green dashed curves represent the flux evolution in a single zone synchrotron and synchrotron self-Compton model with time-dependent injection. For more details about the temporal profile and model curve, the reader is referred to [90]. Reproduced with permission from journal, Reference [90], Copyright 2017, Elsevier.

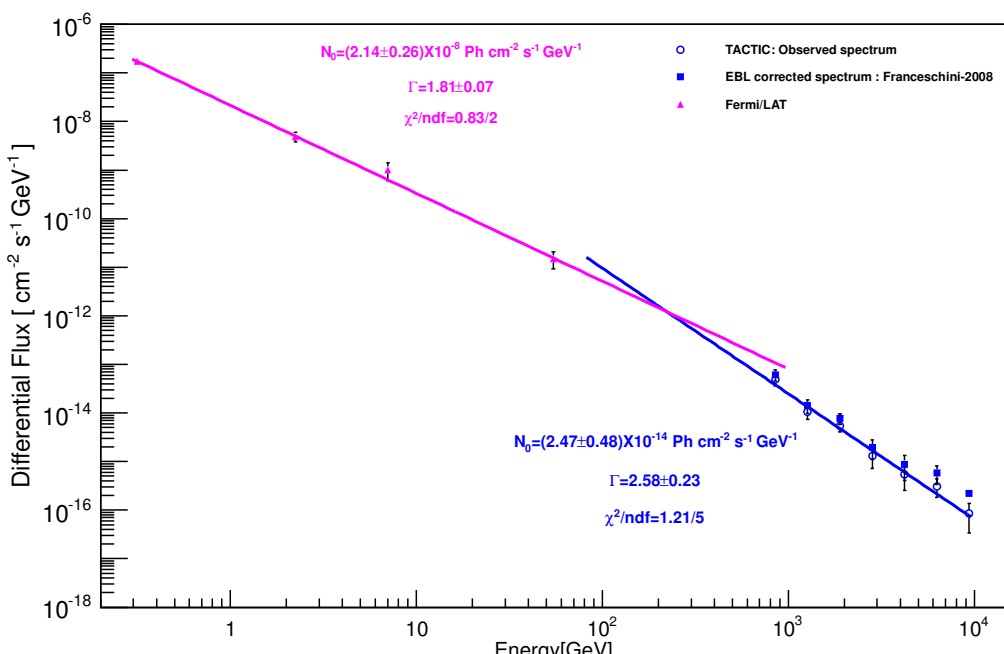

**Figure 5.** $\gamma$-ray spectra of Mrk 421 measured by the *Fermi*-LAT and TACTIC in 0.1–300 GeV and 0.85–9.36 TeV energy bands, respectively, during 15–26 March 2012. Reproduced with permission from journal, Reference [91], Copyright 2017, Elsevier.

### 4.4.3. One Day TeV flare in December 2014

A sudden enhancement in the TeV $\gamma$-ray emission from Mrk 421 was detected by the TACTIC telescope on the night of 28 December 2014 [92]. The TACTIC data on 28 December, 2014 alone resulted in the detection of 86 $\pm$ 17 $\gamma$-ray-like events from Mrk 421 with a statistical significance of 5.17$\sigma$ in an observation time of $\sim$2.2 h above an energy threshold of 0.85 TeV. The high statistics (higher than three Crab Units) of TeV photons enabled us to study the very high-energy $\gamma$-ray emission from the source at shorter timescales. A minimum-variability timescale of $\sim$0.72 days was estimated for the TeV $\gamma$-ray emission during the above flaring episode of the blazar Mrk 421. The integral flux above 0.85 TeV measured with the TACTIC on the night of 28 December, 2014, was estimated to be $(3.68 \pm 0.64) \times 10^{-11}$ ph cm$^{-2}$ s$^{-1}$ in 2.2 h. This was observed to be consistent with the integral flux of $(2.91 \pm 0.0.38) \times 10^{-11}$ ph cm$^{-2}$ s$^{-1}$ above 2 TeV from the near-simultaneous observations with the HAWC observatory for $\sim$6 h on 29 December/2014. This confirmed the sudden increase in TeV $\gamma$-ray activity of Mrk 421 detected by the TACTIC telescope [92]. Near-simultaneous multi-wavelength observations in the one-day broadband spectral energy distribution of the source, shown in Figure 6, were satisfactorily reproduced by simple one-zone leptonic synchrotron and synchrotron self-Compton model. The model parameters estimated from the fitting of the spectral energy distribution were in good agreement with the values reported in the literature for Mrk 421.

Subsequent long-term monitoring of the source with TACTIC during January–February, 2015, for an effective observation time of $\sim$44 h resulted in the detection of 311 $\pm$ 57 $\gamma$-ray photons at a statistical significance of 5.6$\sigma$ [93]. This indicated the low activity state of the source after the sudden flaring episode on 28 December 2014. The time-averaged intrinsic differential energy spectrum in the low-emission state was described by a power law with $f_0 = (1.13 \pm 0.39) \times 10^{-11}$ ph cm$^{-2}$ s$^{-1}$ TeV$^{-1}$ and $\Gamma = 2.34 \pm 0.39$ in the energy range 0.85–15 TeV [93].

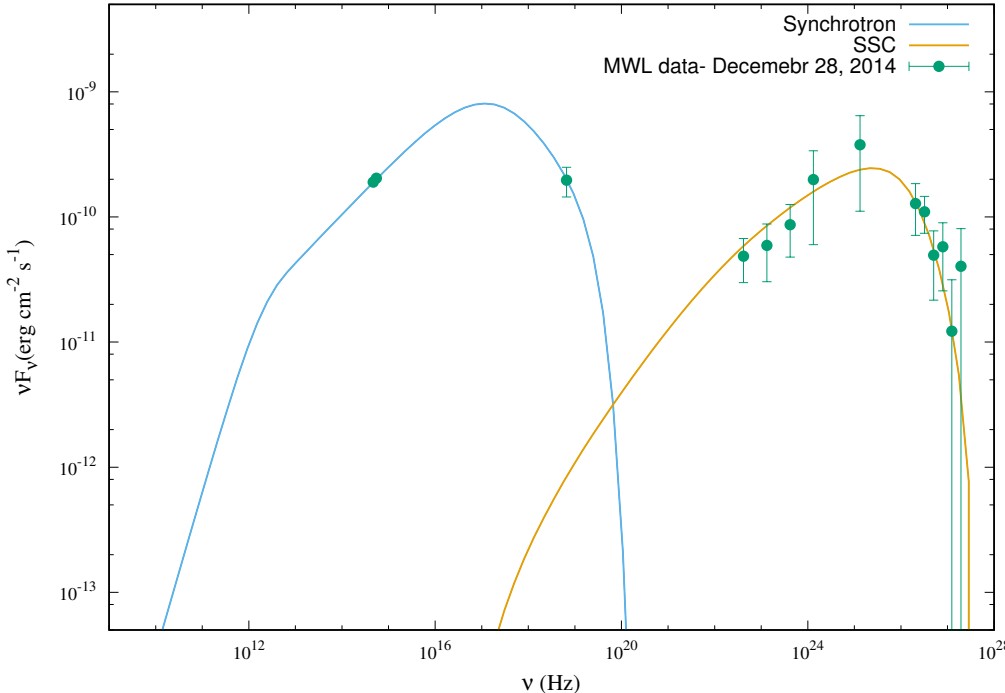

**Figure 6.** Broadband spectral energy distribution of the blazar Mrk 421 under the framework of single-zone homogeneous synchrotron and synchrotron self-Compton (SSC) model observed on 28 December 2014. The multi-wavelength (MWL) data involve near-simultaneous observations from SPOL, Swift, *Fermi*-LAT, and TACTIC. Details of the data set shown in the Figure can be found in [92]. Reproduced with permission from journal, Reference [92], Copyright 2018, Elsevier.

### 4.5. Flare in January 2018

The TACTIC telescope detected an enhanced $\gamma$-ray emission from Mrk 421 on the night of 17 January 2018 [94]. Preliminary analysis of the data collected for 5.6 h indicated that the average flux during this night was around 5 times the Crab nebula flux above 0.85 TeV. The source was also observed on an hourly basis with detection of TeV photons at a significance level of $5\sigma$ and a peak flux of $\sim$7.7 times the Crab nebula flux during the night of 17 January 2018. This was the highest gamma-ray flux recorded by the TACTIC telescope from Mrk 421. This blazar is being regularly monitored by TACTIC as one of potential targets at TeV energies with frequent flaring episodes.

### 4.6. 1ES 1218+304

The blazar 1ES 1218+304 was discovered as an X-ray source at redshift $z = 0.182$. The source was predicted to be a promising candidate for TeV $\gamma$-ray emission from the position of synchrotron peak at X-rays in its broad band spectral energy distribution. With the motivation of detecting TeV gamma-rays, the source 1ES 1218+304 was monitored by the TACTIC telescope during March–April, 2013 for a total observation time of approximately 40 h [95]. No evidence for the TeV $\gamma$-ray emission from the source was found, and therefore a 99% confidence level upper limit on the integral flux above 1 TeV wasestimated as $3.41 \times 10^{-12}$ ph cm$^{-2}$ s$^{-1}$ ($\sim$23% of Crab Nebula flux) assuming a power law differential energy spectrum with $\Gamma = 3.0$ as previously observed by the MAGIC and VERITAS telescopes. Recent long-term multi-wavelength study of 1ES 1218+304 suggested that the source was in steady state with no significant change in its emission activity over a period of 10 years, and optical/UV fluxes were found to be dominated by the host galaxy emission [96]. The stellar emission from the host galaxy was modeled using the PEGASE code. Due to its hard X-ray and TeV $\gamma$-ray spectra, 1ES 1218+304 is an important source for probing the particle acceleration mechanisms that can produce hard power law distribution in the astrophysical jets.



*4.7. B2 0806+35*

B2 0806+35 is a hard spectrum $\gamma$-ray blazar at redshift $z = 0.083$. The first $\gamma$-ray emission from this source was reported by the *Fermi*-LAT in 2010 with an integral flux of $(6.7 \pm 2.5) \times 10^{-10}$ ph cm$^{-2}$ s$^{-1}$ in the energy range 1–100 GeV. The $\gamma$-ray spectra of such sources find special importance in cosmological studies due their intrinsic hard nature. The TACTIC telescope observed B2 0806+35 during December 2015–February 2016 for an effective observations time of $\sim$68 h [97]. Only $79 \pm 45$ $\gamma$-like events with a statistical significance of $1.74\sigma$ were detected by the TACTIC telescope in the direction of B2 0806+35. Therefore, an upper limit of $4.8 \times 10^{-12}$ erg cm$^{-2}$ s$^{-1}$ with $3\sigma$ confidence level on the integral flux above 0.85 TeV was estimated. However, a long-term multi-wavelength study of the source suggested that the $\gamma$-ray spectrum measured by the *Fermi*-LAT is a power law with spectral index of $1.74 \pm 0.15$ in the energy range 0.1–300 GeV [97]. Modeling of the observed UV bump in the broadband spectral energy distribution with a blackbody spectrum indicated a definite trend that the inner radius of the accretion disc was increasing with time whereas the disc temperature was decreasing with time. This suggests that the optically thick and geometrically thin accretion disc was receding from the central black hole [97]. More observations using instruments with higher sensitivity are required to understand the nature of emission from the source.

*4.8. IC 310*

IC 310 is an active galaxy at redshift $z = 0.0189$ located in the Perseus cluster. Radio observations of this galaxy indicate a one-sided core-jet structure blazar-like characteristics. The GeV-TeV emission from this source exhibits substantial flux variability. With the motivation of probing the TeV $\gamma$-ray emission from the jet of such peculiar active galaxy, the TACTIC telescope was used for long-term monitoring of IC 310 from December 2012 to January 2015 for an effective observation time of $\sim$95 h [98]. Only $102 \pm 81$ $\gamma$-ray like events were detected with a statistical significance of $1.26\sigma$ from the source direction. Due to absence of a statistically significant detection in the low activity state, the $3\sigma$ upper limit on the integral flux above 0.85 TeV was as estimated as $4.99 \times 10^{-12}$ ph cm$^{-2}$ s$^{-1}$. This was found to be compatible with the MAGIC telescope results during 2012–2013, when the source was in a low state after a rapid TeV flare on the night of 12–13 November 2012 [99]. The extrapolated flux above 0.85 TeV, calculated using the low activity state energy spectrum derived from MAGIC observations, was obtained to be $2.86 \times 10^{-13}$ ph cm$^{-2}$ s$^{-1}$.

*4.9. NGC 1275*

The radio galaxy NGC 1275 is the brightest member of the Perseus cluster located in its central region at redshift $z = 0.0179$. The TACTIC telescope was deployed to monitor the Perseus galaxy cluster in search of TeV $\gamma$-ray emission since 2012. After the long-term observation of IC 310, it was an obvious choice to monitor the brightest radio galaxy NGC 1275, which is just $\sim$0.6$°$ away from IC 310 in the same galaxy cluster. The TeV observations of NGC 1275 with TACTIC were performed between December 2016 and February 2017 for a live time of $\sim$37 h in the zenith angle range of $20°$–$45°$ [100]. In the absence of a statistically significant detection of TeV $\gamma$-rays from the source direction, the $3\sigma$ upper limit on the integral flux was estimated to be $2.85 \times 10^{-11}$ erg cm$^{-2}$ s$^{-1}$ above 0.85 TeV. In comparison to the other very high-energy observations, the estimated upper limit using the TACTIC telescope was consistent with the low-state-energy spectrum of the source reported by the MAGIC collaboration [101]. NGC 1275 is among a few radio galaxies from which TeV $\gamma$-ray emission has been detected by instruments of better sensitivity. Therefore, this source remains a potential target for the upcoming high-sensitivity MACE telescope.

## 5. Future Roadmap: MACE Telescope

In order to make more effective and significant contributions in the field of GRA and its future science goals, a state-of-the art IACT, MACE (Major Atmospheric Cherenkov Experiment), with improved point source sensitivity and lower threshold energy, has

been recently installed at Hanle (32.8° N, 78.9° E) by the group of Indian gamma-ray astronomers [13]. The altitude of the astronomical site at Hanle (4270 m) in the Himalayan range of North India is the highest for any existing IACT in the world. This highest-altitude Himalayan desert offers an annual average of more than 260 uniformly distributed dark nights, leading to an excellent duty cycle of the telescope for $\gamma$-ray observations. The MACE telescope installed at Hanle site is shown in Figure 7. With an altitude-azimuth mount, the telescope deploys a parabolic light collector of 21 m diameter and 25 m focal length. The light collector comprises 356 mirror panels each of $\sim$1 m $\times$ 1 m size. Each panel consists of indigenously developed 4 diamond turned spherical metallic honeycomb mirror facets of $\sim$0.5 m $\times$ 0.5 m size. This provides a single reflecting surface of area $\sim$346 m$^2$ with uniform reflectivity more than 85%. The mirror facets have graded focal lengths varying from $\sim$25 m to 26.5 m from the centre of light collector to its periphery, . This arrangement of mirrors ensures the minimum on-axis spot size at the focal plane of the telescope. The imaging camera at the focal plane deploys a modular structure with 68 Camera Integrated Modules each having 16 photomultiplier tubes, commonly referred to as pixels. The linear diameter of each photomultiplier tube is 38 mm. All the photomultiplier tubes in the camera are fitted with hexagonal compound parabolic concentrators to cover the entire surface of the camera. The entry apertures of these light concentrators have an angular size of 0.125°. The camera provides a total optical field of view of $\sim$4.36° $\times$ 4.03°. Out of total 1088 pixels, the innermost 576 pixels (36 modules) will be used for event trigger generation with a field of view of $\sim$2.62° $\times$ 3.02° based on a predefined trigger criterion. A trigger configuration of four close-cluster nearest-neighbor pixels is implemented in the MACE hardware. Each 16-channel module has its signal processing electronics built into it. An analogue switched capacitor array DRS-4 operating at $10^9$ Hz is used for continuous digitization of the signal from photomultiplier tubes.

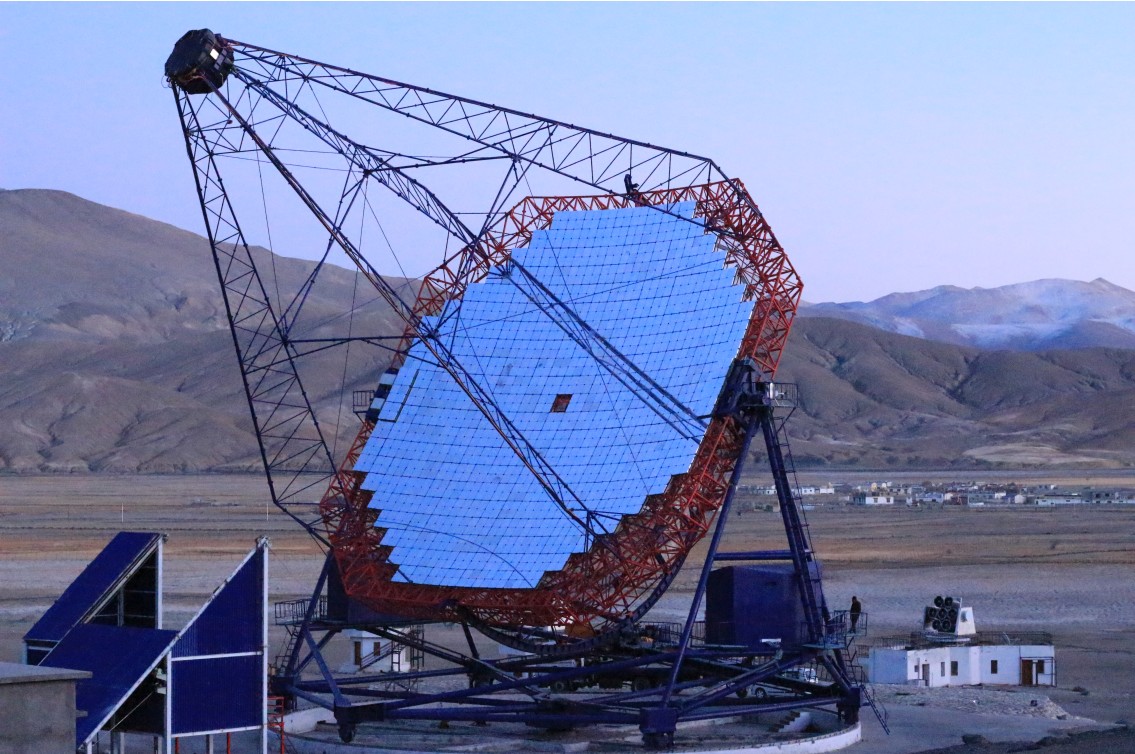

**Figure 7.** The 21 m diameter MACE $\gamma$-ray telescope at Hanle site (4270 m above sea level).

*5.1. Expected Performance of MACE*

The simulation study of the trigger performance using *CORSIKA* package [102] suggests that the $\gamma$-ray trigger energy threshold of the MACE telescope is $\sim$20 GeV in the low zenith angle range of 0°–40° and increases to $\sim$173 GeV for large zenith angle of 60° [103].

As expected, for any IACT, integral rates for MACE are dominated by protons with nearly 80% contribution to total trigger rate. In the zenith angle range of $0°$–$40°$, the integral rate is estimated to be $\sim$650 Hz, and it decreases sharply to $\sim$305 Hz at $60°$ zenith angle. The 50 h integral sensitivity of the telescope is estimated to be $\sim$2.7% of the Crab Unit at the analysis energy threshold of $\sim$38 GeV at $5°$ zenith angle [104]. This has been estimated by carrying out the $\gamma$-hadron segregation using the Random Forest method. A comparison of the integral sensitivity of the MACE telescope with MAGIC-I is shown in Figure 8. It is evident from Figure 8 that, compared to the MAGIC-I telescope, the MACE telescope has a lower analysis energy threshold (as expected on account of higher altitude). Furthermore, it is observed that the MACE telescope would be more sensitive than the MAGIC-I telescope up to $\sim$150 GeV energy. The estimated energy and angular resolutions of the MACE telescope as a function energy are reported in Figure 9. The telescope is expected to have an energy resolution of $\sim$40% in the energy range of 30–47 GeV, which improves to $\sim$20% in the energy bin 1.8–3 TeV [105]. Furthermore, the angular resolution of the MACE telescope is estimated to be $\sim$0.21° in the energy range of 30–47 GeV, and it improves to a value of $\sim$0.06° in the energy range of 1.8–3 TeV [105]. Overall expected performance of MACE is similar to that MAGIC-I, although MACE has lower analysis energy threshold $\sim$30 GeV. The MACE telescope is expected to see its first light in March–April 2021.

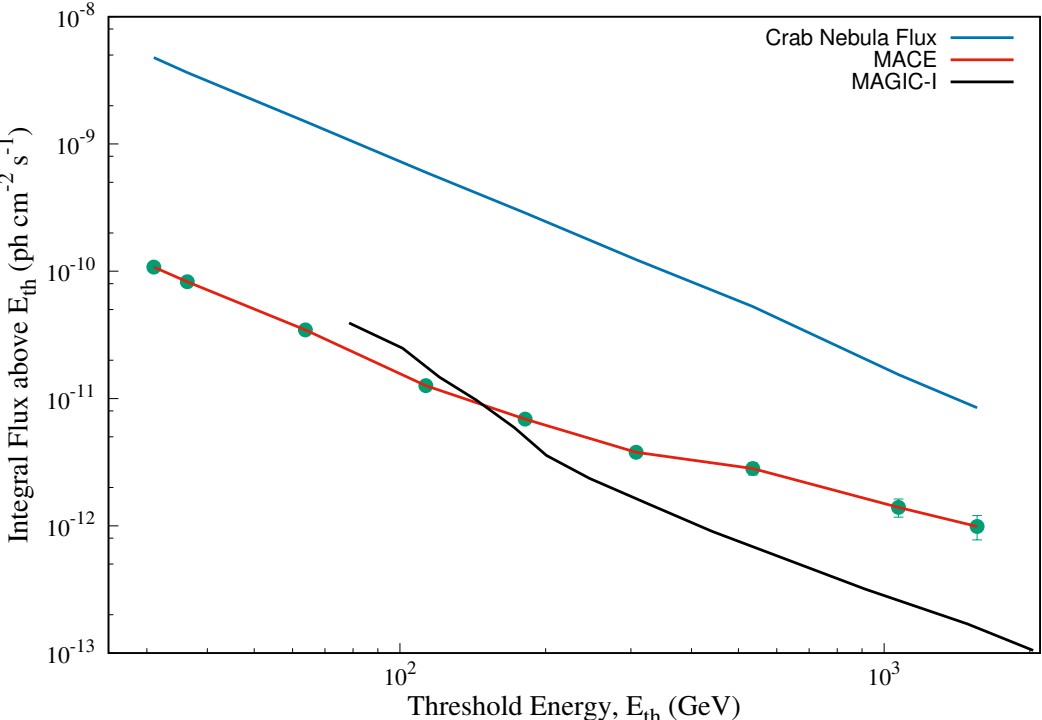

**Figure 8.** Integral flux sensitivity of the MACE telescope and its comparison with that of MAGIC-I [105].

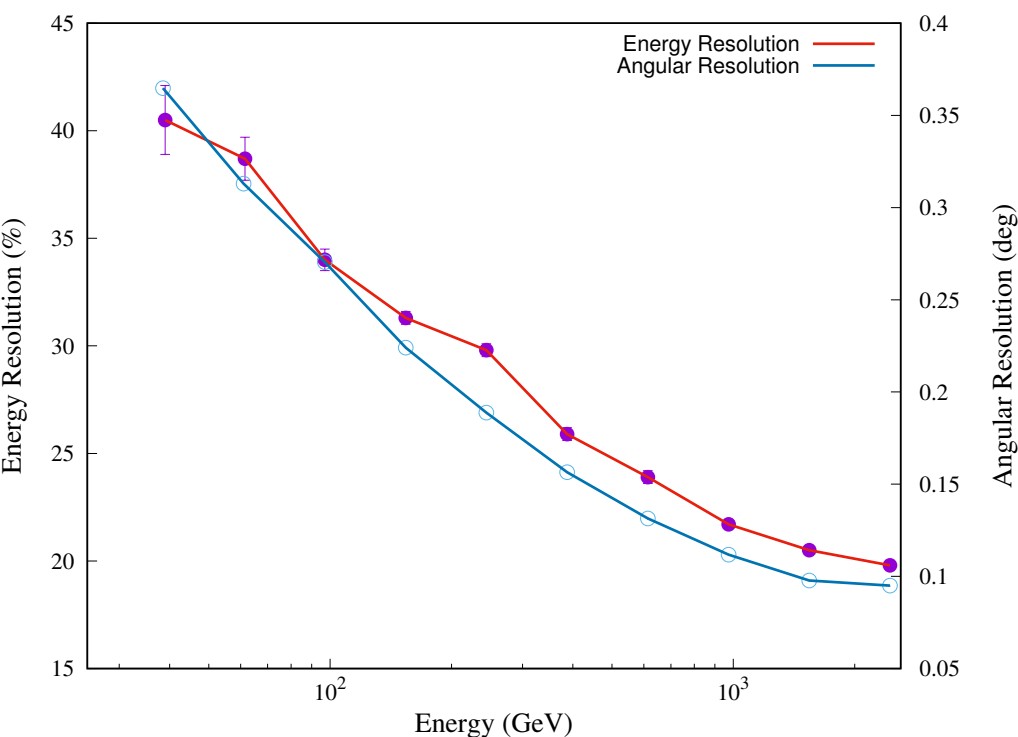

**Figure 9.** Energy and angular resolution of the MACE γ-ray telescope as a function of energy [105].

*5.2. MACE on the World Map*

On the world map, the MACE telescope fills up the longitudinal gap between different major IACTs (H.E.S.S., MAGIC, and VERITAS) operating around the globe. The altitude of MACE site (4270 m) makes it the highest altitude IACT in the world. Given the construction of CTA (Cherenkov Telescope Array) observatory (https://www.cta-observatory.org (accessed on 8 February 2021)) both in northern and southern hemispheres, the MACE telescope (21 m diameter) also has the distinction of being second largest IACT after the Large Size Telescope (23 m diameter) in the northern hemisphere and third largest IACT after H.E.S.S.-II (28 m diameter) in the world. Therefore, along with other existing IACTs, MACE will be very useful for exploring the GeV-TeV γ-ray sky, in particular for continuous monitoring campaigns of flaring sources. The MACE telescope, having an excellent energy overlap with the space-based γ-ray detectors like *Fermi*-LAT [106], will be extremely important to explore several outstanding problems in GRA over the next decade. The *Fermi*-LAT fourth source catalog (4FGL) has reported more than 5000 γ-ray sources including 3130 blazars and 239 pulsars above $4\sigma$ significance [107]. Although *Fermi*-LAT has detection capability beyond 500 GeV, its sensitivity above 100 GeV is not sufficient to detect weak γ-ray emission from cosmic sources. Therefore, there is a strong possibility to detect γ-rays from more sources with IACTs like MACE having better sensitivity in the tens of GeV energy regime and for exploring the exact nature of unidentified sources reported in the 4FGL catalog. The lower energy threshold of MACE is expected to help in observation of the deep γ-ray Universe beyond redshift $z > 2$ and monitoring of astrophysical transients like gamma ray bursts. On the other hand, MACE, with an analysis energy threshold of ~30 GeV, will play an important role in the study of pulsars, which are assumed to have a separate GeV-TeV emission component. Apart from exploring the Universe at GeV-TeV energies, MACE has the capability to address a range of open problems in observational cosmology such as constraining the density of EBL photons and strength of intergalactic magnetic field, probing the nature of dark matter candidates like weakly interacting massive particles of masses above 1 TeV, the existence of axion-like particles beyond the standard model of particle physics, and so on.

## 6. Conclusions

Ground-based GRA in India has been pursued for the last five decades. The activity started with low-sensitivity instruments at different locations in India. However, the journey of Indian GRA using IACTs over the last two decades seems to be very satisfying on the whole. It has experienced an era of very encouraging and impressive advances at both scientific and technological fronts. Starting from the 81-pixel prototype camera of the TACTIC telescope to the development of 1088-pixel camera for the MACE telescope (second largest IACT at the highest altitude in the northern hemisphere), indicates the advances made in the field of technology and resources in a very short span of time. At present, the TACTIC telescope is among the few ground-based telescopes in the world being used for observations of TeV $\gamma$-ray emission from different astrophysical sources. The telescope provides a unique opportunity to monitor potential $\gamma$-ray sources in the multi-TeV energy regime during flaring and low activity states. Despite relatively limited sensitivity, the TACTIC telescope has provided very important information during the flaring episodes of blazars like Mrk 421 and Mrk 501 in the TeV energy range. The energy spectra of TeV $\gamma$-ray photons derived using TACTIC observations are very useful in understanding the relativistic acceleration of charged particles in astrophysical environments. Upper limits obtained from the non-detections of the TeV $\gamma$-ray emission from the different sources after a long monitoring with the TACTIC telescope are important in constraining the source emission models for production of $\gamma$-ray photons in the TeV regime. The upper limit reported for a source observed with the TACTIC telescope can also be used for constraining the variability properties if the source has been previously detected or will be detected at a later time during the flaring state. Such observations are also important in obtaining the information from all the past observations from various telescopes to make predictions for future instruments. The MACE telescope with lower energy threshold and higher sensitivity is expected to provide path-breaking results in GRA over the next decade.

**Author Contributions:** Conceptualization, K.K.S. and K.K.Y.; Writing–original draft preparation, K.K.S.; Supervision, K.K.Y. All authors have read and agreed to the published version of the manuscript.

**Funding:** This research received no external funding.

**Institutional Review Board Statement:** Not applicable.

**Informed Consent Statement:** Not applicable.

**Acknowledgments:** Authors thank the anonymous reviewers for their critical comments and suggestions. We are grateful to all the former and present colleagues of Astrophysical Sciences Division at Bhabha Atomic Research Centre, Mumbai for their contributions in this article. A special thank to our colleague Chinmay Borwankar for sharing the MACE simulation results presented in this article.

**Conflicts of Interest:** The authors declare no conflict of interest.

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
