# Peer review of "20 Years of Indian Gamma Ray Astronomy Using Imaging Cherenkov Telescopes and Road Ahead"

_universe, doi:10.3390/universe7040096_

Round 1
Reviewer 1 Report
Overall assessment:
The paper presents a historical review on the development of gamma-ray telescopes in India. The contribution of the Indian scientists to this field is very high which justifies the main proposal of the paper. The paper does not present new research results nor offer a particular view of the historical development. The way it is written, the paper offers a partial compilation of previously published results without adding any historical or scientific analysis. The paper contains a list of facts and dates rather than a comprehensive description and interpretation of the great achievements done in India. It is expected that in this sort of review paper, the authors analyse the highlights showing how they connect to each other drawing the timeline of the research in the country as well as how they compare to other developments in other countries. Therefore I do not recommend the publication of this paper before a major revision. Below I list some aspects which I believe would help to improve the paper.
Introduction: The introduction lists some milestones in the development of gamma-ray astronomy. It is composed of two long paragraphs which mixes space- and ground-bases experiments, theoretical and experimental developments, international and Indian aspects. Despite I could not find any mistake in the text, I believe the reader would benefit from an introduction more focused on the development of Indian experimental activities. The way it is written, the introduction does not work as an invitation to the reader to continue with the next sections. I would suggest to segment the introduction in several paragraphs, focus the main message in the subject of the paper (development of IACTs in India) and after that make short connections with other aspects (space-based, theory, international development etc).
Section 2: “Evolution of Indian GRA with IACT”. This section is the paper therefore I don’t understand why it is in a separate section. Maybe this section should be “Early times” and focus on the birth of the area in India.
Section 3: “TATIC Telescope”. I believe the reader would benefit from comments and analysis on how TATIC compares to other contemporary experiments. What are the strong and weak points of the detector ? What are the improvements in data analysis ?
line 38: "Early observations using atmospheric Cherenkov technique were challenged by the dominant contribution of Cherenkov light from the air showers initiated by cosmic ray protons.": Seems out of context, does not connect to the previous or next sentences.
line 92: The use of reference 20 to independently establish the presence of the knee-feature in the cosmic-ray spectrum is debatable. Please offer more evidence and more references or re-write the sentence.
line 130: "close to Davies-Cotton design". Please specify. Is it possible to tell the differences from a Davies-Cotton design.
line 139: cite CORSIKA
Section 4: “Important Results from TACTIC Observations”. I believe the readers would benefit from an analysis of the importance of the results. The results are described and almost all subsections finish with an observation of the need for extra observations. I believe it would be great to have a short description of the impact of each result.
line 371: “Therefore, this source remains a potential target for upcoming high sensitivity telescopes like MACE and CTA (Cherenkov Telescope Array).”: This sentence seems to imply that MACE and CTA have equal sensitivity.
Section 5: Future Roadmap: MACE has a science case. It is going to be the only large dish IACT to run above 4000 m above sea level. A short of the telescope configuration is given in this section. A very short description (less than 5 lines) of the performance it given. Almost nothing about the science impact is written despite the overlap with Fermi-LAT. I believe the reader would be interested in knowing the scientific impact of MACE.
line 401: cite Fermi-LAT
In section 5 or 6, the authors should describe how MACE fits in the international scenario given the construction of CTA.
Reviewer 2 Report
- This paper gives a nice review of India's gamma-ray astronomy development, a detailed summary of the TACTIC telescope's results, and a short section about the new MACE telescope that'll start taking data soon. In my opinion, it is suitable for publication with only a few things to add.
- In Section 5 about the MACE telescope, the paper provides many detailed technical details and a nice picture of MACE, but I think some plots on its performance potential will be more useful.
- Also, a plot showing the comparison of MACE's expected performance and other telescopes (HESS, MAGIC, VERITAS, CTA) is needed, without such comparison it's hard to see MACE's place in the world.
- Section 5 mentions the angular resolutions of MACE in the text, a plot of angular resolution vs. energy would be good (to accompany the text).
Round 2
Reviewer 1 Report
The authors have reviewed the paper taking into consideration my first report. Some sentences were corrected and some section ammended. I have no further suggestions.
Author Response
We thank the reviewer for accepting the revised manuscript.